# Cryo-EM structure revealed a novel F-actin binding motif in a *Legionella pneumophila* lysine fatty acyltransferase

Wenjie W Zeng[1,2], Garrison Komaniecki[3†], Jiaze Liu[2‡], Hening Lin[3,4], Yuxin Mao[1,2]*

[1]Weill Institute for Cell and Molecular Biology, Cornell University, Ithaca, United States; [2]Department of Molecular Biology and Genetics, Cornell University, Ithaca, United States; [3]Department of Chemistry, Department of Molecular Biology and Genetics, Cornell University, Ithaca, United States; [4]Howard Hughes Medical Institute; Department of Medicine and Department of Chemistry, The University of Chicago, Chicago, United States

*For correspondence:
ym253@cornell.edu

Present address: †Chemistry Department, University of Wisconsin – Parkside, Kenosha, United States; ‡Department of Pharmacology, Yale University School of Medicine, New Haven, United States

Competing interest: The authors declare that no competing interests exist.

## eLife Assessment

This **important** study by Zeng et al characterizes a novel Legionella pneumophila effector, Llfat1 (Lpg1387), which binds actin through a newly identified actin-binding domain. Data is **convincing**; structural analysis of the Llfat1 ABD-F-actin complex enabled the development of this domain as a probe for F-actin. Additionally, the authors show that Llfat1 functions as a lysine fatty acyltransferase targeting small GTPases, highlighting its importance in both bacterial pathogenesis and cytoskeletal biology.

**Abstract** *Legionella pneumophila* is an opportunistic bacterial pathogen that causes Legionnaires' disease. To establish an intracellular niche conducive to replication, *L. pneumophila* translocates a diverse array of effector proteins that manipulate various host cellular processes, including the actin cytoskeleton. In a screen for effectors that alter actin dynamics, we identified a *Legionella* effector, Lfat1 (lpg1387), which colocalizes with the actin cytoskeleton in eukaryotic cells. Lfat1 specifically binds F-actin through a novel actin-binding domain (ABD). High-resolution cryo-electron microscopy (Cryo-EM) analysis revealed that this ABD forms a long α-helix hairpin, with its tip interacting with subdomains I and II of two adjacent actin molecules within the F-actin filament. Interestingly, while individual α-helices of the hairpin fail to bind F-actin, co-expression as separate fusion proteins restores binding activity. Furthermore, we demonstrated that Lfat1 exhibits lysine fatty acyltransferase (KFAT) activity, targeting host small GTPases. These findings establish a foundation for studying the KFAT family of bacterial toxins and uncover a novel F-actin-binding motif, providing an alternative F-actin marker with notable flexibility.

## Introduction

The actin cytoskeleton plays an essential role in diverse cellular processes, including cell motility, cytokinesis, intracellular trafficking, and cell signaling (*Dominguez and Holmes, 2011*; *Letort et al., 2015*; *Pollard, 2016*). Actin is one of the most conserved, ubiquitous, and abundant proteins in cells from amoebas to humans (*Pollard, 2016*). Actin exists in two distinct forms: the monomeric G-actin form and the double-stranded filamentous F-actin form. F-actin is highly dynamic with a net association of ATP-actin to the barbed (+) end and dissociation of ADP-actin monomers from the pointed (-) end (*Pollard, 2016*). The assembly and disassembly of F-actin in vivo is intricately regulated through

interactions with a structurally and functionally diverse family of actin-binding proteins (ABPs) (*Pollard, 2016*). These ABPs have usually been classified according to their functional properties into several families, such as the actin monomer-binding protein profilin (*Carlsson et al., 1977*), actin nucleators, including Arp2/3 and Formin that initiate de novo branched and unbranched filament assembly, respectively (*Chen et al., 2010*; *Dong et al., 2003*; *Machesky et al., 1994*; *Pizarro-Cerdá et al., 2017*), the heterodimeric capping proteins that terminate F-actin elongation (*Isenberg et al., 1980*), severing/depolymerization factors Cofilin and Gelsolin (*Bamburg et al., 1980*; *Barrie et al., 2025*; *Tanaka et al., 2018*; *Yin and Stossel, 1979*), filament binding proteins, such as tropomyosin that binds to the main groove of an actin filament and stabilizes the filament (*von der Ecken et al., 2015*; *Yu and Ono, 2006*), and cross-linking proteins that crosslink and stabilize multiple F-actin filaments together for cell movement and muscle contraction (*Le et al., 2017*; *Ribeiro et al., 2014*). The accumulation of three-dimensional (3D) structures of ABPs in complex with actin revealed that different ABPs share a limited number of actin-binding structural modules (*Van Troys et al., 1999*). Thus, identifying and characterizing new actin-binding structural modules will provide direct hints of the actin target site and the functional effect on the actin dynamics of ABPs that share the specific actin-binding module.

Given actin's essential roles in eukaryotes, many prokaryotic and viral pathogens co-opt a variety of mechanisms that target the host actin cytoskeleton for effective pathogenesis (*Aktories et al., 2011*; *Bugalhão et al., 2015*). The virulence factor BimA from *Burkholderia pseudomallei* mimics host actin-polymerizing proteins Ena/VASP to nucleate, elongate, and bundle filaments (*Benanti et al., 2015*). The *Vibrio parahaemolyticus* VopL consists of a VopL C-terminal domain and three WASP homology 2 motifs and mimics the Arp2/3 complex and formin proteins to stimulate actin polymerization (*Namgoong et al., 2011*; *Zahm et al., 2013*). The *Salmonella* invasion protein A effector is an ABP that enhances actin polymerization and promotes the uptake efficiency of the bacterium (*McGhie et al., 2004*).

The Gram-negative bacterium *Legionella pneumophila* is the causative agent of a potentially fatal form of pneumonia in humans named Legionnaires' disease (*Cunha et al., 2016*; *Fraser et al., 1977*; *McDade et al., 1977*; *Mondino et al., 2020*). Upon entry into human alveolar macrophage cells, the facultative intracellular pathogen translocates more than 350 different bacterial proteins, known as effectors (*Burstein et al., 2009*; *Huang et al., 2011*; *Zhu et al., 2011*). These effector proteins subvert multiple conserved eukaryotic pathways, such as ubiquitination (*Price and Abu Kwaik, 2021*; *Tomaskovic et al., 2022*), autophagy (*Choy et al., 2012*; *Omotade and Roy, 2020*; *Thomas et al., 2020*; *Wan et al., 2024*), lipid metabolism (*Hsu et al., 2012*; *Swart and Hilbi, 2020*; *Toulabi et al., 2013*), and the actin cytoskeleton (*Franco et al., 2012*; *Prashar et al., 2018*; *Zhang et al., 2023*) to aid the pathogen in establishing a *Legionella*-containing vacuole amenable to intracellular growth and proliferation (*Gomez-Valero et al., 2019*; *Mondino et al., 2020*; *Oliva et al., 2018*).

Like other bacterial pathogens, *L. pneumophila* utilizes a cohort of virulent effectors to modulate actin. Recent studies revealed that the VipA effector nucleates actin and disrupts the multivesicular bodies pathway (*Franco et al., 2012*) and the RavK effector cleaves actin to abolish actin polymerization (*Liu et al., 2017*). In our recent screen for *L. pneumophila* effectors that affect host F-actin dynamics, we identified several novel effector proteins that exhibited various degrees of F-actin-associated phenotypes. Among these positive hits, Lpg1387, an effector with no known function, showed strong colocalization with F-actin. In this study, we report the identification of a novel actin-binding motif consisting of a long antiparallel α-helical hairpin. We further revealed the molecular mechanism of actin-binding by cryo-electron microscopy (Cryo-EM) and presented evidence for developing a potential F-actin probe based on this novel actin-binding motif. Moreover, using click chemistry, we confirmed that, in addition to the actin-binding motif, Lpg1387 has a lysine fatty acylate (KFA) catalytic domain specific for small GTPases. Hence, we named this *L. pneumophila* effector Lfat1 (*Legionella F-actin-binding fatty-acyl-transferase 1*).

## Results

### The *Legionella* effector Lfat1 directly interacts with F-actin via a coiled-coil domain

To explore how the intracellular bacterial pathogen *L. pneumophila* modulates host actin dynamics, we performed a screen to search for effectors that perturb host actin structures. In this screen, we

imaged F-actin structures with phalloidin staining in HeLa cells transfected with a GFP-effectors library. In this screen, several effectors showed various degrees of F-actin-associated phenotypes (*Figure 1—figure supplement 1*). Among the positive hits, MavH (Lpg2425) has recently been shown to polymerize actin filaments in a membrane-dependent manner (*Zhang et al., 2023*). Another effector, Lfat1 (Lpg1387), exhibited nearly a complete colocalization with F-actin (*Figure 1—figure supplement 1*).

To elucidate the molecular mechanism of how Lfat1 localizes to F-actin filaments, we first analyzed the 3D structure predicted by AlphaFold (*Jumper et al., 2021*). The structure revealed a hammer-like structure for the full-length Lfat1 protein (*Figure 1A*). The head of the hammer is formed by a globular NC-domain (N- and C-terminal globular domain), which is contributed by both the N-terminal (residues 1–137, red) and the C-terminal lobes (residues 356–469, pink) of the protein, while the handle of the hammer is formed by an elongated, antiparallel, coiled-coil hairpin (CC-domain), which contains the middle portion of the protein (residues 138–355, cyan). To map the region responsible for Lfat1 F-actin localization, we created constructs expressing the NC- and CC-domains fused with an N-terminal GFP, respectively, and investigated their intracellular localization by fluorescence microscopy. Interestingly, while the NC-domain showed a diffused cytosolic localization, the CC-domain exhibited a high colocalization to actin filaments comparable to the wild-type (WT) protein (*Figure 1B and C*). This result was further confirmed by an immunoprecipitation (IP) experiment wherein full-length Lfat1 and the CC-domain were able to pull down actin, whereas the NC-domain could not (*Figure 1D*). To test whether the CC-domain directly binds actin, we performed an in vitro F-actin co-sedimentation assay, in which purified recombinant proteins of the CC-domain were incubated with actin in the presence of G-actin or F-actin buffer. Following ultracentrifugation to pellet F-actin, the supernatant and pellet were analyzed on SDS-PAGE (*Figure 1E*). In the G-actin buffer, both actin and the CC-domain protein remained in the supernatant; however, the CC-domain protein co-sedimented with the polymerized F-actin in the F-actin buffer, indicating that the CC-domain of Lfat1 directly binds to F-actin. Strikingly, when the CC-domain protein was incubated with actin at a 1:1, 1:2, or 1:4 (actin: CC) molar ratio in the F-actin buffer, an approximately equal amount of CC-domain proteins was co-sedimented with F-actin, and the excess CC-domain proteins remained in the supernatant. This observation indicates that the interaction between the CC-domain and actin is saturable, and the binding occurs at a one-to-one molar ratio (*Figure 1F*). Together, our findings identified Lfat1 as a novel actin-binding effector of *Legionella*. We further demonstrated that Lfat1 binds F-actin at a one-to-one stoichiometry through a unique, long coiled-coil hairpin CC-domain, which we will henceforth call the actin-binding domain (ABD).

## Cryo-EM structure of the Lfat1 ABD-F-actin complex

The discovery of a novel ABD triggered us to interrogate the molecular mechanism of actin-binding by this ABD. We sought to determine the Cryo-EM structure of the Lfat1 ABD-F-actin complex. In our initial attempts to prepare the protein complex, F-actin bundles were readily induced by Lfat1 ABD with the F-actin buffer, making it hard to solve a single F-actin filament for structural determination (data not shown). To restrict excessive actin polymerization, equal molar of Lfat1 ABD and G-actin were incubated in a non-polymerizing G-actin buffer overnight at 4°C. The protein complex sample was then applied to cryo-grids, vitrified, and loaded to a 200 kV Thermo Fisher Talos Arctica transmission electron microscope for Cryo-EM data collection. The data were processed, and a high-resolution density map (average to 3.5 Å resolution) was calculated and refined using CryoSPARC (*Punjani et al., 2017*). The atomic model of the complex was built by docking the F-actin structure (PDB: 7BTI) and the AlphaFold-predicted model of Lfat1 ABD into the Cryo-EM density using ChimeraX (*Meng et al., 2023*). The model was then refined iteratively using Phenix (*Liebschner et al., 2019*), and the final model was validated online by Worldwide Protein Data Bank (wwPDB) validation server at https://validate.wwpdb.org; (*Figure 2—figure supplement 1* and *Table 1*).

The Cryo-EM density map of the complex allowed a complete resolution of the actin subunit in the F-actin filament, including its bound ADP and $Mg^{2+}$ ion (*Figure 2A*, *Figure 2—figure supplement 1A and B*). The D-loop (DNase I-binding loop) of the actin monomer adopts a closed conformation. The D-loop of the nth actin monomer extends into the hydrophobic cleft between actin subdomains 1 and 3 of the n+2nd actin monomer. The hydrophobic residues (V45, M46, V47, and M49) at the tip of the D-loop pack against a large hydrophobic area lining the wall of the hydrophobic cleft of the n+2nd actin monomer (*Figure 2B*, *Figure 2—figure supplement 2C*). The D-loop also mediates specific

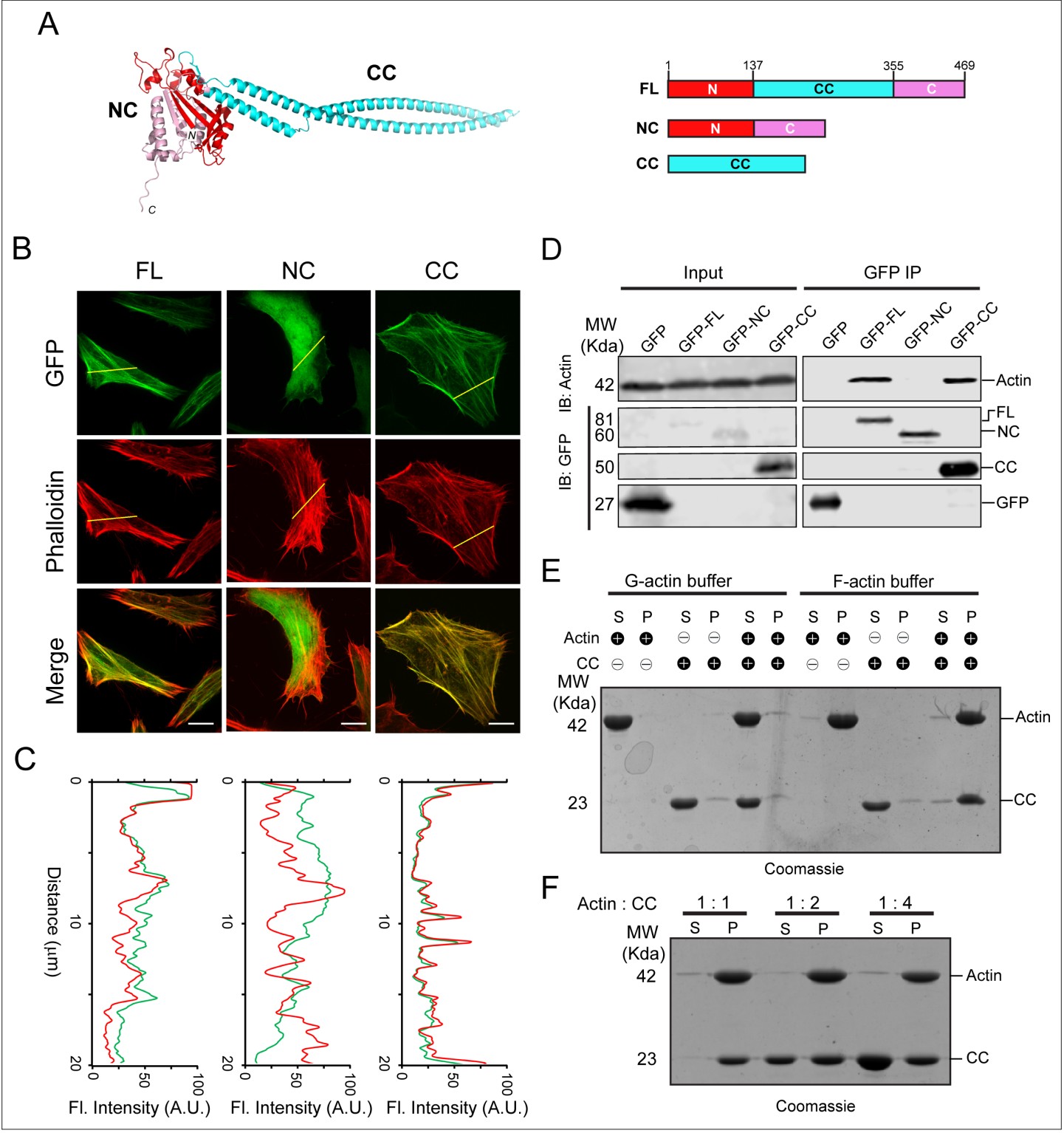

**Figure 1.** Identification of LFat1 (lpg1387) as an F-actin binding effector. (**A**) AlphaFold-predicted structure of Lfat1 (left) and domain architecture of Lfat1 (right) with the N-terminal domain shown in red, the C-terminal domain in pink, and the central coiled-coil domain in cyan. FL: full-length, NC: N and C globular domain, CC: coiled-coil domain, respectively. (**B**) Cellular localization of Lfat1-FL, -NC, or -CC as determined by fluorescence microscopy. HeLa cells transiently expressing GFP-fused Lfat1-FL, -NC, or -CC were fixed and stained with phalloidin. Scale bar = 10 μm. (**C**) Colocalization was determined by fluorescence intensity line scan along the yellow line shown in (**B**). Red = F-actin, green = GFP. (**D**) Co-immunoprecipitation to determine interaction of Lfat1 with actin. HEK293T cells transiently expressing either GFP-empty vector, -Lfat1 FL, -Lfat1 NC, or -Lfat1 CC were lysed, and cell lysates were immunoprecipitated using anti-GFP nanobeads. The IP samples were analyzed with SDS-PAGE followed by immunoblot against GFP and

*Figure 1 continued on next page*

*Figure 1 continued*

actin. (**E**) Co-sedimentation assay to determine direct interaction between Lfat1 CC and F-actin. Purified G-actin, CC, or G-actin plus CC was incubated either in G-actin buffer or F-actin polymerization buffer, then ultracentrifuged to separate supernatant from pellet, followed by analysis via SDS-PAGE. S: supernatant, P: pellet. (**F**) Binding stoichiometry between Lfat1 CC and actin as determined by co-sedimentation assay.

The online version of this article includes the following source data and figure supplement(s) for figure 1:

**Source data 1.** Original western blots and SDS-PAGE Coomassie staining gels displayed in *Figure 1D, E, and F*.

**Source data 2.** PDF files of original western blots and SDS-PAGE Coomassie staining gel displayed in *Figure 1D, E, and F* with labels.

**Figure supplement 1.** Intracellular localization of representative *Legionella* effectors.

**Table 1.** CryoEM Data collection, refinement and validation statistics.

| | |
|---|---|
| Microscope | FEI Talos Artica |
| Voltage (keV) | 200 |
| Defocus range (um) | –0.4 to –3.0 |
| Camera | K3 direct electron detector |
| Pixel size (Å) | 0.833 (super-resolution) |
| Total electron dose (e/Å2) | 40.68 |
| Exposure time (seconds) | 1.23 |
| Frames per movie | 50 |
| Number of images | 4548 |
| **3-D refinement statistics and helical symmetry** | |
| Total number of particles | 1,220,462 |
| Resolution (Å) | 3.58 |
| Helical twist | –167 |
| Rise | 28 |
| **Model composition and validation** | |
| Non-hydrogen atoms | 34,740 |
| Protein residues | 4390 |
| Ligands | 10 Mg, 10ADP |
| **RMSD** | |
| Bond lengths(Å) | 0.25 |
| Bond angles (°) | 0.5 |
| B-factor (Å2) Protein | 58.87 |
| B-factor (Å2) Ligand (ADP) | 56.6 |
| MolProbity Score | 1.9 |
| Clashscore | 5 |
| Ramachandran plot:-Favored | 95 |
| Allowed | 5 |
| Outlier | 0 |
| PDB ID | 8VAA |
| EMDB Code | 43087 |

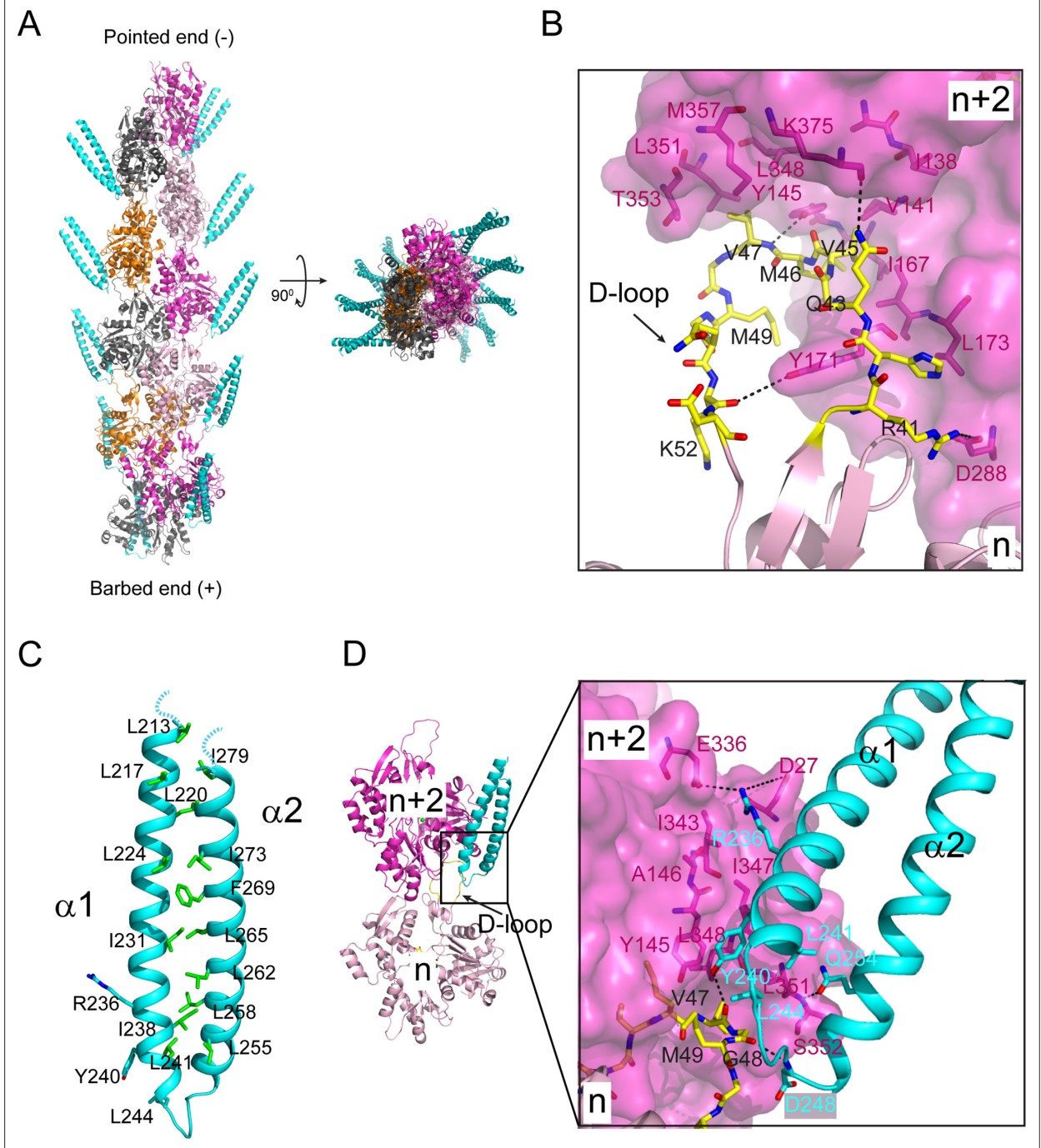

**Figure 2.** Cryo-electron microscopy (Cryo-EM) structure of the Lfat1 ABD in complex with F-actin. (**A**) Cryo-EM structure of the F-actin-Lfat1 ABD complex. Left: Side view of the structure positioned with the pointed end (-) up and barbed end (+) down. The visible part of ABD is colored in cyan. Right: Top view of the complex. (**B**) The D-loop conformation and its interactions in the hydrophobic cleft of the n+2nd actin monomer. (**C**) Ribbon diagram of the distal portion of the Lfat1 coiled-coil domain. The two α-helices are zipped together through extensive hydrophobic interactions contributed mainly by leucine and isoleucine residues (shown in sticks). (**D**) Structural representation of the interaction between the ABD domain of Lfat1 and F-actin. Inset: Extensive hydrophobic, hydrogen bonding, and electrostatic interactions were observed between the Lfat1 ABD domain and the two adjacent actin monomers (for details, see in the text).

The online version of this article includes the following figure supplement(s) for figure 2:

**Figure supplement 1.** Cryo-electron microscopy (Cryo-EM) data processing details of F-actin-Lfat1ABD complex.

**Figure supplement 2.** Cryo-electron microscopy (Cryo-EM) maps of the F-actin-Lfat1ABD complex.

**Figure supplement 3.** The interface between Lfat1 ABD and F-actin.

hydrogen bond interactions between the two adjacent actin monomers. The main chain amino group of V47 and the carbonyl group of K52 of the D-loop hydrogen bond with the hydroxyl groups of Y145 and Y171, respectively (*Figure 2B*).

The Cryo-EM structure also revealed the distal portion of the coiled-coil hairpin, which consists of about 1/3 of the entire Lfat1 ABD domain (*Figure 2—figure supplement 2D*). The proximal end of the ABD domain is not visible, likely due to its flexibility and zeroed out in 2D-class averaging. The structure revealed that the two α-helices of the Lfat1 ABD domain are zipped together by a stretch of hydrophobic residues, mostly leucines and isoleucines (*Figure 2C*). The ABD domain radiates away from the central F-actin core with its tip of the hairpin region binding to the site between two adjacent actin molecules within each strand of the F-actin filament (*Figure 2A*). The ABD domain embeds a surface area of 3703 $Å^2$ on the actin filament, which is contributed by both the D-loop region of the nth and the hydrophobic cleft of the n+2nd actin monomers (*Figure 2—figure supplement 3*). Several hydrophobic residues (Y240, L241, and L244) located at the tip of the ABD hairpin are accommodated by a hydrophobic pocket formed between the two adjacent actin molecules (*Figure 2D*). The interaction between the ABD domain and the actin filament also involves several hydrogen bonds. The hydroxyl group of Y240 of the ABD domain forms a hydrogen bond with the main chain carbonyl group of V47 at the D-loop; the amino group of D248 of the ABD pairs with the carbonyl oxygen of G48; and the side chain carbonyl oxygen of ABD Q254 makes hydrogen bond with the main chain amine group of S352 of actin. Moreover, salt bridges are also observed between R236 of the ABD domain and D27 and E336 of the n+2nd actin monomer (*Figure 2D*).

Together, our Cryo-EM structure of the Lfat1 ABD domain in complex with F-actin revealed the intricate molecular basis of multivalent interactions between F-actin and a novel prokaryotic ABD. In addition, the complex structure revealed a 1:1 ratio of the interaction between the Lfat1 ABD and actin monomer, which is in agreement with the stoichiometry determined by the previous co-sedimentation experiment (*Figure 2A*, *Figure 2—figure supplement 2C and D*).

## Validation of key residues on the ABD domain in its recognition of F-actin

To validate our structural observations of the interaction between Lfat1 ABD and F-actin, we performed an alanine substitution mutagenesis experiment of three representative residues (R236, Y240, and Q254) in the Lfat1 ABD domain (*Figure 3A*). We found that GFP-tagged R236A or Q254A ABD mutant showed a slight increase in diffused signals, with the majority of proteins remaining colocalized with F-actin. However, the Y240A mutation renders the protein mostly cytosolic (*Figure 3B and C*). To further validate the fluorescence imaging results, we performed an in vitro F-actin co-sedimentation titrating assay to measure the binding affinity between ABD proteins and F-actin (*Figure 3D and E*). The apparent $K_d$ for WT ABD to F-actin is calculated at about 1.48 µM, which is on par with LifeAct ($K_d$ of 2.2 µM) (*Riedl et al., 2008*). Consistently, the $K_d$ for R236A and Q254A mutants increased about 10-fold, 19.06 and 16.28 µM, respectively. More strikingly, the Y240A mutant showed a substantial decrease in affinity with a $K_d$ of 63.69 µM (*Figure 3D and E*). In summary, the mutagenesis experiments confirmed that multivalent interactions contribute to the binding of Lfat1 ABD with F-actin, with hydrophobic interactions playing a central role, and the affinity and specificity were further enhanced by hydrogen bond and salt bridge interactions.

## Comparison of Lfat1 ABD with other ABDs

The discovery of a novel ABD from the *Legionella* effector Lfat1 prompted us to compare this unique ABD to other representative ABPs. Although ABDs adopt a diverse structural fold, most of them share a similar interaction scheme with F-actin by targeting a hotspot encompassing the D-Loop of Actin$_n$ and the hydrophobic cleft of Actin$_{n+2}$ (*Figure 4A and B*; *Dominguez, 2004*). For example, LifeAct, which is derived from the first 17 residues from *Saccharomyces cerevisiae* ABP140 (*Riedl et al., 2008*), utilizes the hydrophobic residues aligned on one side of its amphipathic α-helix to engage primarily hydrophobic interactions with a small hydrophobic patch at the F-actin hotspot (*Belyy et al., 2020*; *Kumari et al., 2020*). The actin-binding CH1 domain of Utrophin (a neuromuscular junction scaffolding protein) contains multiple F-actin-binding sites (*Keep, 2000*; *Keep et al., 1999*). Two of the actin-binding sites on the CH1 domain interact primarily with the D-loop region of the nth actin monomer, and the third one, consisting of the N-terminal α-helix, spills the interface further into the subdomain

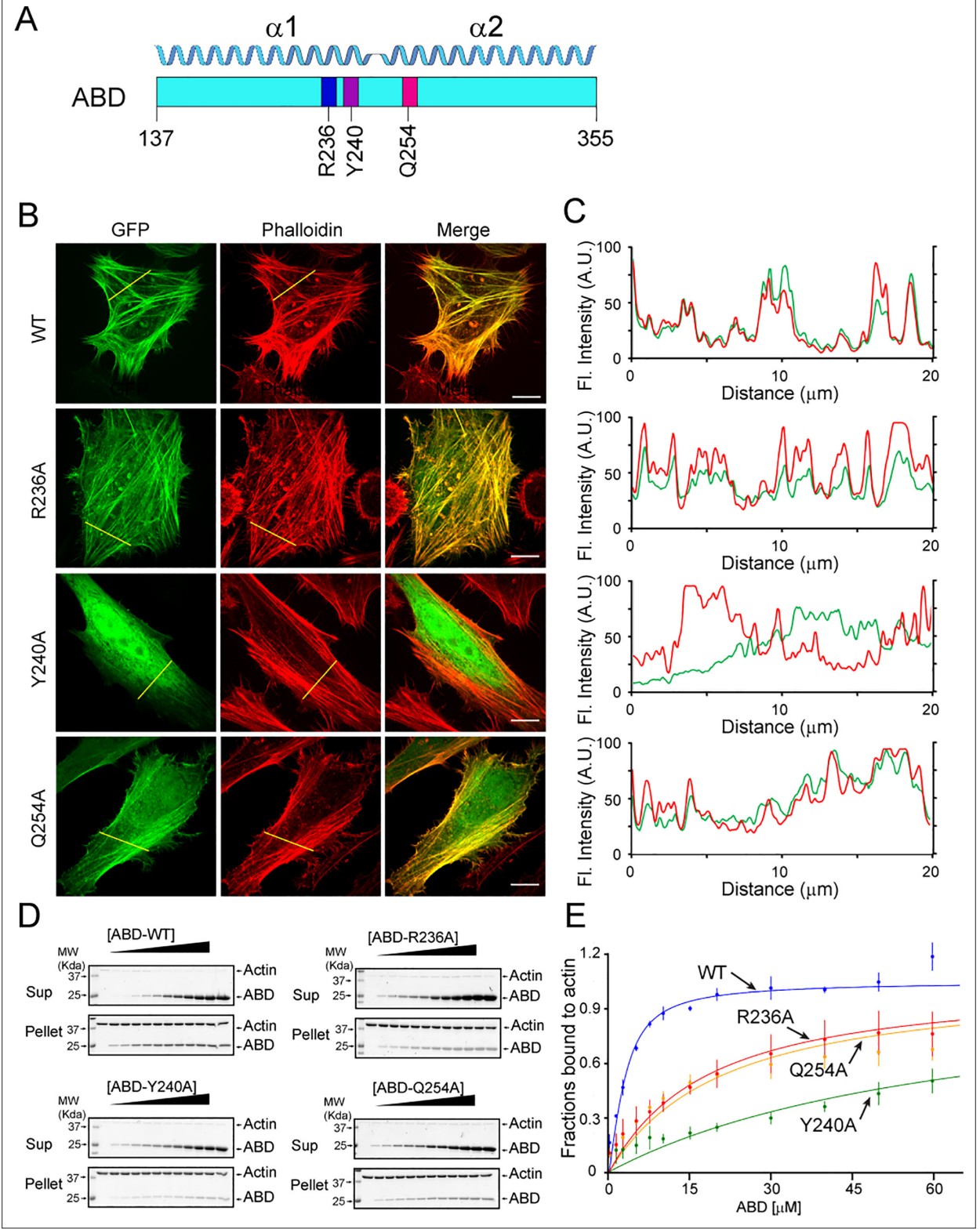

**Figure 3.** Validation of key Lfat1 ABD residues in their contributions to F-actin interactions. (**A**) Schematic diagram of the Lfat1 ABD domain. Three key residues (R236, Y240, and Q254) involved in actin binding are labeled. (**B and C**) F-actin localization analysis of indicated ABD mutants. GFP-Lfat1 WT, R236A, Y240A, or Q254A mutant was transiently expressed in HeLa cells followed by fixation and staining with phalloidin. Fluorescence images were taken by a confocal microscope and analyzed using line scan along the indicated yellow lines. Scale bar = 10 μm. (**D**) Co-sedimentation assay of F-actin with wild-type (WT) and mutant Lfat1 ABD. Increasing amounts (0–60 μM) of recombinant WT or mutant ABD proteins were incubated with a fixed

*Figure 3 continued on next page*

*Figure 3 continued*

amount of actin. The samples were ultracentrifuged after 30 min of room temperature incubation in 1× actin polymerization buffer. The supernatant and pellet fractions were analyzed by SDS-PAGE. (**E**) Quantitative analysis of the co-sedimentation titration data. The data point for each concentration was averaged from three technical replicates. The error bar represents the standard deviation.

The online version of this article includes the following source data for figure 3:

**Source data 1.** Original SDS-PAGE Coomassie staining gel displayed in *Figure 3D*.

**Source data 2.** PDF files of original SDS-PAGE Coomassie staining gels displayed in *Figure 3D* with labels.

I region of the nth actin subunit (*Kumari et al., 2020*). The *Pseudomonas aeruginosa* effector protein, Exotoxin Y or ExoY, uses its C-terminal 'anchor' helix to engage primarily hydrophobic interactions with the hydrophobic cleft in subdomain 1 of the n+2nd actin subunit (*Belyy et al., 2021*) in a way similar to that of LifeAct (*Belyy et al., 2020*). ExoY also contains a peptide, which meanders on the surface of the n+2nd actin subunit and functions as a 'sensor' for the actin activator but contributes little affinity to F-actin binding (*Belyy et al., 2021*). These examples support that the hydrophobic cleft formed by subdomains 1 and 3 is the 'hotspot' for many ABDs, and the binding site on actin is frequently extended to the vicinity of the 'hotspot' depending on unique features associated with each ABD (*Figure 4A*).

A structural comparison of the ABD-F-actin complex structures also revealed that the actin D-loop adopts a variety of conformations upon the binding of different ABDs (*Figure 4B*). In globular G-actin, the D-loop is either disordered or adopts an α-helix, depending on its nucleotide state or binding with G-actin-binding proteins (*Graceffa and Dominguez, 2003*). In F-actin, the D-loop inserts itself into the hydrophobic target-binding cleft of the n+2 subunit immediately above it (*Das et al., 2020*; *Dominguez and Holmes, 2011*). The D-loop region is also involved in direct interactions with many ABDs (*Figure 4*). In the Lfat1 ABD and F-actin complex, specific hydrogen bonds are formed between the backbone carbonyl group of D-loop residues V47 and G48 and the ABD residues Y240 and D248 (*Figure 2D*). These hydrogen bonding interactions cause the D-loop to insert slightly deeper into the hydrophobic cleft and induce a unique conformation of the D-loop residues G50 and Q51 (*Bos taurus* numbering, equivalent to G48 and Q49 in *Gallus gallus*) not observed in other structures (*Figure 4B*). This observation suggests that although the hydrophobicity of the 'hotspot' plays a dominant role in ABD binding, the capacity to accommodate the large variety of actin-binding motifs at the 'hotspot' is likely due to the structural plasticity of the D-loop.

## Engineering novel F-actin probes derived from the Lfat1 ABD

Fluorescent toxins or proteins are frequently used as F-actin probes in fixed or live cells; however, they all have certain limitations (*Belin et al., 2014*; *Courtemanche et al., 2016*; *Lemieux et al., 2014*; *Munsie et al., 2009*). The discovery of a new F-actin-binding domain from the *Legionella* effector Lfat1 inspired us to investigate whether it can be developed as an alternative in vivo F-actin probe. We first tried to map the minimum F-actin binding region in Lfat1 ABD. A series of ABD truncations: ABD-S1 (residues 171–323), ABD-S2 (190–306), and ABD-S3 (211–280) were created and transiently expressed in HeLa cells. All three truncated versions demonstrated specific colocalization with F-actin comparable to the full-length ABD (*Figure 5A*). However, further shortening of this ABD resulted in loss of function and exhibited a complete cytosolic location (data not shown). Thus, our studies revealed a novel F-actin probe consisting of a 70 amino acid-long α-helix hairpin.

The Lfat1 ABD contains two long α-helices forming a hairpin. We next asked whether this ABD remains functional if the α-helix hairpin is split into two individual α-helices. To test this, we fused an N-terminal mCherry with α1 and a C-terminal GFP with the α2 of the ABD, respectively, and examined their intracellular localization. All these single α-helix fusions showed a diffused localization (*Figure 5—figure supplement 1A and B*); however, to our surprise, when the two fusion constructs encoding full-length ABD-α1 and α2 were expressed together, these two α-helices were able to form a functional ABD and colocalized with F-actin as the intact WT ABD (*Figure 5C and D*). Interestingly, the α-helices derived from ABD-S1 could also reconstitute a functional F-actin probe, but not the further shortened α-helices derived from ABD-S2 (*Figure 5C and D*). These results suggested that the F-actin probe derived from the Lfat1 ABD can be used in a split form, providing flexibility to this new probe.

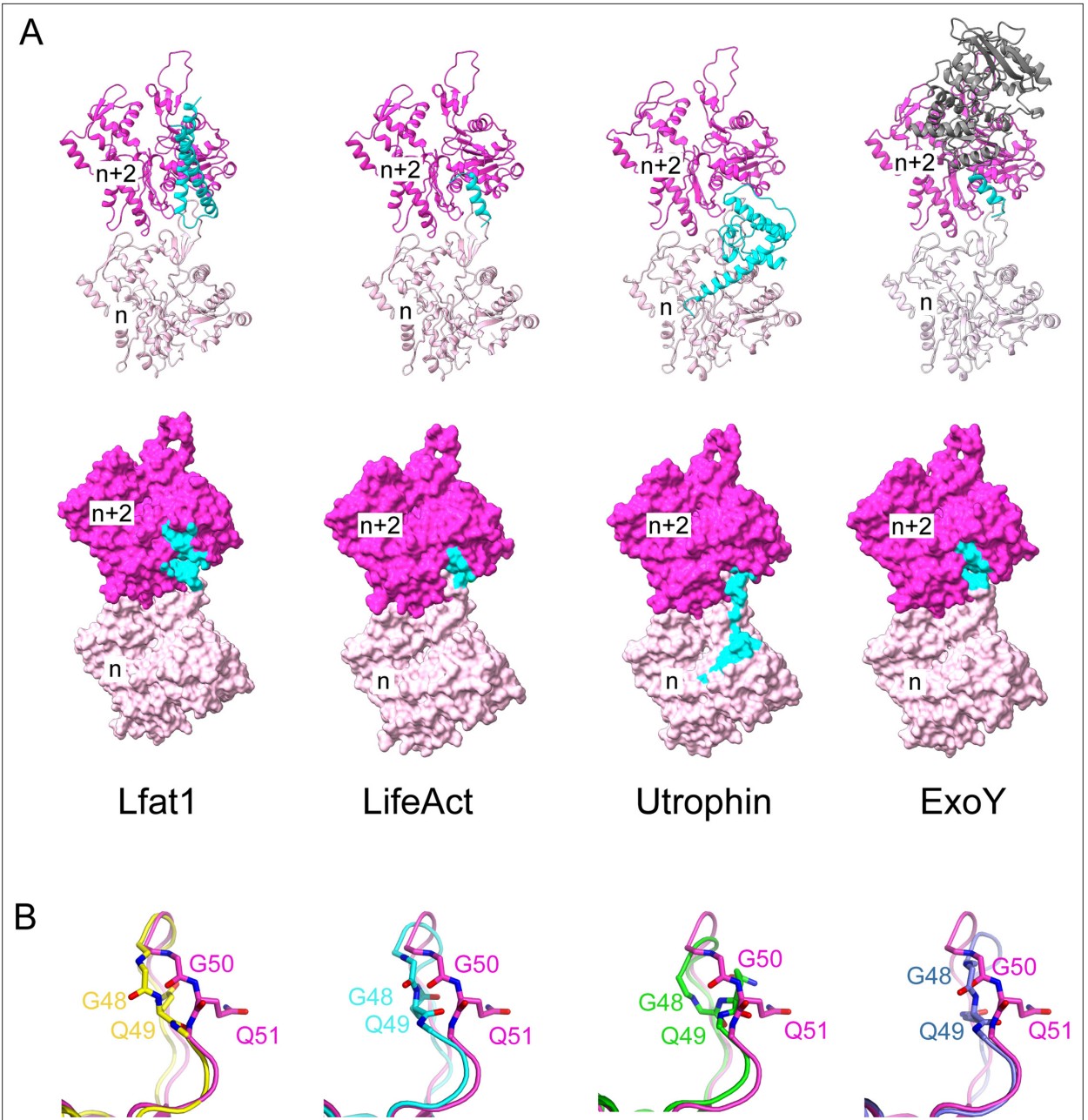

**Figure 4.** Structural comparison between Lfat1 and other ABD-F-actin complexes. (**A**) Ribbon diagram (upper row) and surface representation (bottom row) of two adjacent actin monomers (purple and pink) bound with Lfat1 ABD, LifeAct (PDB ID: 7BTE), Utrophin (6M5G), and ExoY (7P1G). The region interfacing with each ABD is colored in cyan. (**B**) Structural comparison of the D-loop conformation in the Lfat1 ABD-F-actin complex (purple) with the D-loop in other F-actin-ABD complexes: F-actin alone (yellow), LifeAct (cyan), Utrophin (green), and ExoY (navy blue). Two D-loop residues with the largest deviation of backbone dihedral angles (G50 and Q51) are shown in sticks.

## Lfat1 is a lysine fatty acyltransferase

The AlphaFold-predicted structure of Lfat1 revealed a globular domain composed of the N- and C-terminus beside the central, coiled-coil hairpin (*Figure 1A*). Structural homology search using the DALI server (*Holm and Rosenström, 2010*) yielded the top hit as the RID (Rho GTPase Inactivation Domain) toxin from the *Vibrio vulnificus* (PDB:5XN7) with a Z-score of 7.9. The catalytic domain of both proteins contains a central β-sheet flanked by multiple α-helices. The conserved catalytic dyad (H38 and C403) in Lfat1 was positioned with a similar orientation to the dyad in the RID toxin (H2595 and C2835) (*Figure 6A*).

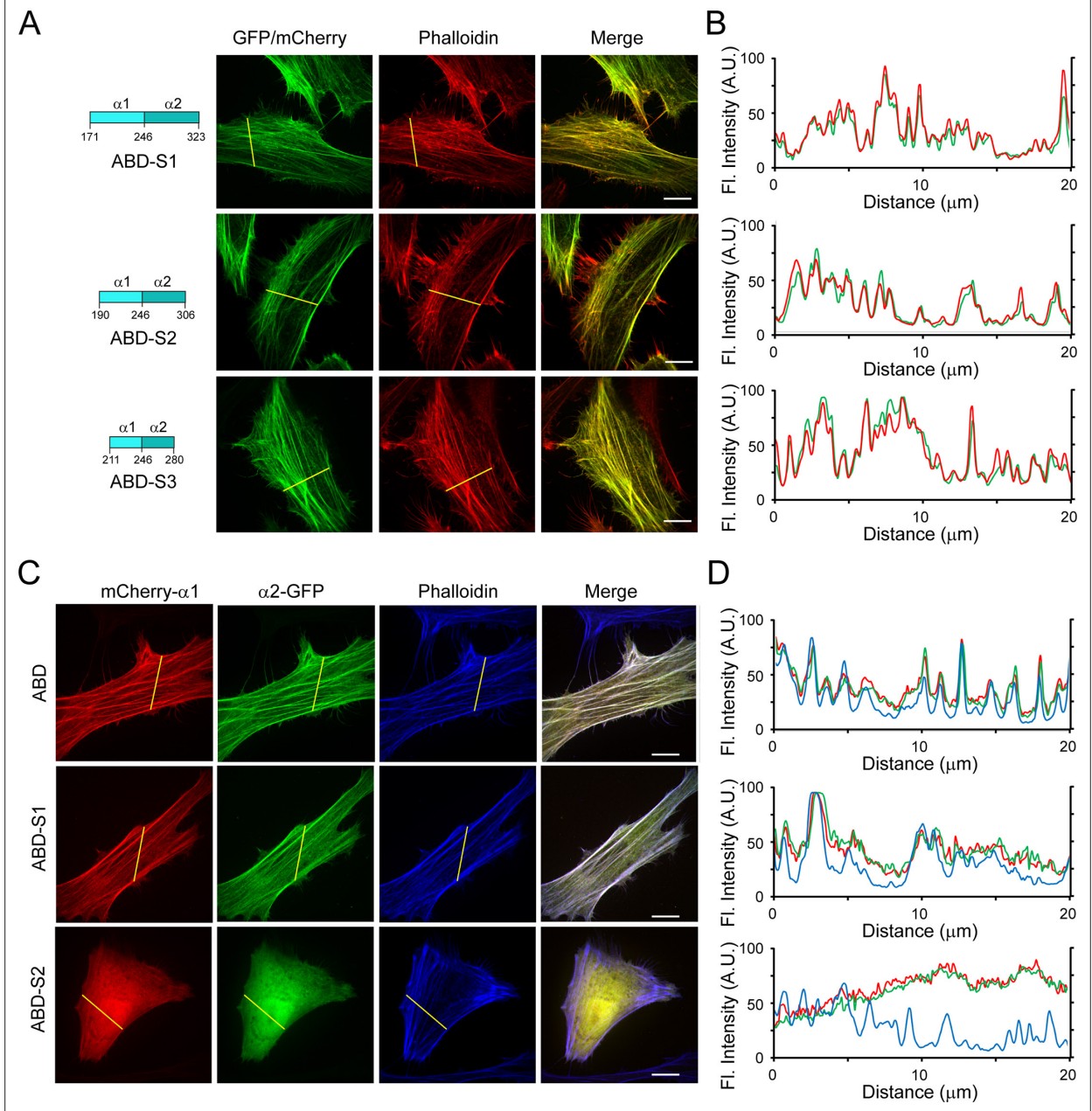

**Figure 5.** Engineering Lfat1 actin-binding domain (ABD) as an in vivo F-actin probe. (**A**) Mapping the minimal ABD of Lfat1. Schematic of shortened Lfat1ABD fragments used for F-actin binding (Left). Representative fluorescence images of cells expressing indicated ABD fragments and stained with rhodamine-phalloidin. (**B**) Line-scan analysis for the indicated ABD probes along the yellow lines. (**C**) Representative fluorescence images of cells transiently transfected with plasmids expressing separated α-helices (mCherry-α1 and α2-GFP) of ABD, ABD-S1, and ABD-S2. The cells were fixed and stained with CF647-phalloidin. (**D**) Line-scan analysis of the images shown in (**C**). Scale bar = 10 μm.

The online version of this article includes the following figure supplement(s) for figure 5:

**Figure supplement 1.** Intracellular localization of separated individual α-helices of the Lfat1 actin-binding domain (ABD).

The RID toxin is a lysine fatty acyltransferase (KFAT) processed from a much larger prototoxin that transfers long acyl chains to the ε-amine of lysines in small GTPases, such as those in the Rac subfamily (**Zhou et al., 2017**). To determine whether Lfat1 possesses a similar KFAT activity, we probed the fatty acylation of a small GTPase Rac3. Briefly, HEK-293T cells were co-transfected with plasmids expressing Flag-Rac3 and GFP-Lfat1 or its catalytic mutants followed by treatment with Alk14, a clickable chemical analog of palmitic acid modified with a terminal alkyne moiety. Flag-Rac3 proteins were enriched

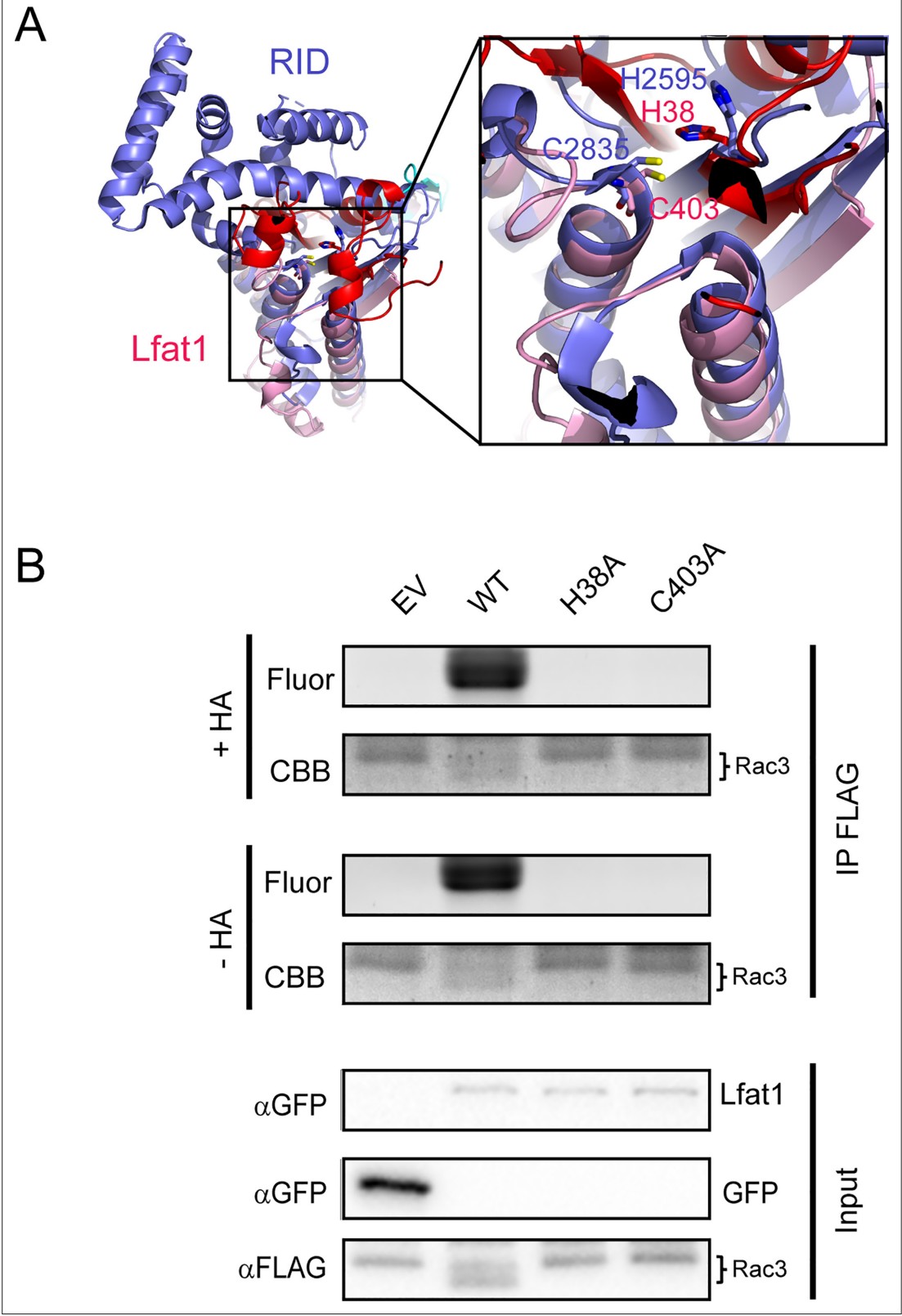

**Figure 6.** Lfat1 is a lysine fatty acyltransferase that modifies eukaryotic small GTPases. (**A**) Ribbon representation of the *V. vulnificus* Rho Inactivation Domain (RID) (PDB ID: 5XN7) catalytic domain (purple) superimposed with the AlphaFold-predicted NC-domain of Lfat1 (N-terminal domain in red, C-terminal domain in pink). Inset: the catalytic pockets of the two proteins with conservation of the catalytic histidine and cysteine between RID and Lfat1 shown in sticks. (**B**) Lfat1 catalyzes lysine fatty acylation of Rac3. N-terminal Flag-tagged Rac3 was co-expressed with either GFP empty vector (EV),

*Figure 6 continued on next page*

*Figure 6 continued*

GFP-Lfat1 WT, H38A, or C403A mutant in HEK293T cells for 24 hr. Flag-Rac3 was enriched using immunoprecipitation and subjected to click chemistry. The samples were then separated on SDS-PAGE and scanned for fluorescence signals.

The online version of this article includes the following source data and figure supplement(s) for figure 6:

**Source data 1.** Original western blots/gels corresponding to *Figure 6B*.

**Source data 2.** PDF files of original western blots/gels corresponding to *Figure 6B* with labels.

**Figure supplement 1.** Identification of potential Lfat1 substrates by click chemistry-coupled Stable Isotope Labeling by Amino Acids in Cell Culture (SILAC) mass spectrometry.

**Figure supplement 1—source data 1.** Original western blots/gels corresponding to *Figure 6—figure supplement 1C, D, E, and F*.

**Figure supplement 1—source data 2.** PDF files of original western blots/gels corresponding to *Figure 6—figure supplement 1C, D, E, and F* with labels.

**Figure supplement 2.** The lysine fatty acyltransferase activity of Lfat1 does not depend on actin binding.

**Figure supplement 2—source data 1.** Original western blots/gels corresponding to *Figure 6—figure supplement 2*.

**Figure supplement 2—source data 2.** PDF files of original western blots/gels corresponding to *Figure 6—figure supplement 2*.

from transfected cells by IP and were then conjugated to an azide-containing fluorophore, TAMRA-$N_3$, via copper-catalyzed cycloaddition (*Hein et al., 2008*). The reaction products were separated by SDS-PAGE and analyzed by in-gel fluorescence detection and western blot. Robust fluorescence signals associated with Rac3 were detected in the presence of WT Lfat1, but not its H38A or C403A catalytic mutants (*Figure 6B*). Furthermore, we found that Rac3-associated fluorescence signals were stable even after the hydroxylamine (HA) treatment, which can reverse acylation on cysteine but not lysine residues. We next asked whether Lfat1 can fattyacylate other host targets. To address this question, we performed a click chemistry-coupled SILAC (Stable Isotope Labeling by Amino Acids in Cell Culture) mass spectrometry (MS) experiment to identify potential targets (*Figure 6—figure supplement 1A*). Interestingly, nearly half of the top hits are small GTPases, including Rabs, RheB, RalA, and Rap1B (*Figure 6—figure supplement 1B*). Fatty acylation by Lfat1 on several of the small GTPases was further verified using click chemistry (*Figure 6—figure supplement 1C–F*). Furthermore, the fatty acylation of small GTPases by Lfat1 does not appear to depend on F-actin binding, as these substrates were still modified by the actin-binding-deficient Lfat1 Y240A mutant (*Figure 6—figure supplement 2A and B*). Together, our results demonstrated that the *Legionella* effector Lfat1 is a bona fide KFAT that potentially fattyacylates host small GTPases when exogenously expressed in cultured cells.

## Discussion

In this study, we identified a novel ABD from *L. pneumophila*. This prokaryote-originated ABD has an α-helical hairpin-like structure, which is unprecedented from any other known ABDs. We further mapped the minimum region (~70 residues) required for actin binding and demonstrated the feasibility of using this prokaryote ABD as an alternative F-actin probe. Another unique and potentially useful characteristic of this probe is its ability to function as a split-ABD. Although the individual α-helix derived from the Lfat1 ABD fails to bind actin, they were able to reconstitute a functional intact ABD when co-expressed. This unique feature can be harnessed to target multicomponent biological complexes to F-actin by genetically fusing individual components to the split α-helices. Furthermore, our Cryo-EM structure revealed that the Lfat1 ABD intersects with F-actin obliquely. Its N- and C-termini are pointed away with an adjustable distance from the filament depending on the size of the designed ABD. Thus, the Lfat1 ABD can be used to target proteins of interest to F-actin with a tunable distance from the filament to achieve spatial distribution-related specificity.

F-actin probes are essential tools in cell biology for visualizing and studying the dynamics of the actin cytoskeleton in living and fixed cells. These probes come in various forms, including fluorescently labeled phalloidins, ABPs, and genetically encoded fluorescent actin markers (*Melak et al., 2017*). Each type of probe has its advantages and limitations. The choice of probe depends on the specific experimental requirements, such as whether live-cell imaging is needed, the level of perturbation that can be tolerated, and the ease of use. The toxic chemical derived from fungi, phalloidin, has been developed as the gold standard F-actin marker to stain actin in fixed samples and tissues (*Cooper, 1987*). However, it is not suitable for live-cell imaging due to its toxicity and low cell permeability.

Many yeast- or human-derived ABDs have been developed into F-actin probes by fusion with fluorescent proteins. LifeAct (*Riedl et al., 2008*), Utrophin (*Burkel et al., 2007*), and F-tractin (*Brehm et al., 2004*) are the three most commonly used genetically encoded probes. Although these probes have been widely used for live-cell imaging, they suffer from problems such as low affinity for F-actin and perturbation in actin dynamics. Our discovery of a novel ABD offers an alternative F-actin probe that not only can be used to study actin dynamics but also can be used as a versatile anchor to target specific activities to F-actin.

The discovery of Lfat1 as an F-actin-binding KFAT raised the intriguing question of whether its enzymatic activity depends on F-actin binding. Recent studies have shown that other *Legionella* effectors, such as LnaB and Ceg14, use actin as a co-factor to regulate their activities. For instance, LnaB binds monomeric G-actin to enhance its phosphoryl-AMPylase activity toward phosphorylated residues, resulting in unique ADPylation modifications in host proteins (*Fu et al., 2024*; *Wang et al., 2024*). Similarly, Ceg14 is activated by host actin to convert ATP and dATP into adenosine and deoxyadenosine monophosphate, thereby modulating ATP levels in *L. pneumophila*-infected cells (*He et al., 2025*). However, this does not appear to be the case for Lfat1. We found that Lfat1 mutants defective in F-actin binding retained the ability to modify host small GTPases when expressed in cells (*Figure 6—figure supplement 2*). These findings suggest that, rather than serving as a co-factor, F-actin may serve to localize Lfat1 via its ABD, thereby confining its activity to regions enriched in F-actin and enabling spatial specificity in the modification of host targets.

Our finding that Lfat1 is a protein KFAT provides important insights to understand the physiological function of Lfat1. The RID is a module found in Multifunctional-Autoprocessing Repeats-in-Toxin (MARTX) toxins produced by certain Gram-negative pathogenic bacteria, such as *Vibrio cholerae* and *V. vulnificus* (*Satchell, 2015*). RID primarily targets the Rho GTPase family members by the covalent attachment of long-chain fatty acids to the ε-amino groups of lysine residues. The modification inactivates the small GTPases, leading to disruption of the actin cytoskeleton and consequent cell rounding and hence facilitating bacterial invasion and impairing host immune responses (*Zhou et al., 2017*). A recent study reported that RID modifies other host proteins, notably septins. Fatty acylation on septins alters the localization and compromises the host cell structural integrity (*Xu et al., 2024*). In this study, we demonstrated that the globular NC-domain of Lfat1 exhibits KFAT activity when overexpressed in vivo. Using click chemistry, we showed that Lfat1 could fatty acylate lysines of the host small GTPase Rac3 (*Figure 6B*) and other small GTPases (*Figure 6—figure supplement 1*). Many of the small GTPase substrates we identified are known to associate with and regulate actin. For example, RhoG regulates the actin cytoskeleton in lymphocytes (*Vigorito et al., 2003*). Rap1 is reported to regulate actin reorganization and microtubule organizing center polarization at the B cell immune synapse (*Wang et al., 2017*), RheB is reported to regulate actin filament distribution (*Gau et al., 2005*), Ral GTPases (RalA and RalB) link Ras, Rac, Rho signaling to control cell migration (*Zago et al., 2019*), and RAB8A regulates spindle migration via ROCK-mediated actin assembly in mouse oocyte meiosis (*Pan et al., 2019*). The identification of the KFAT effectors among all *Legionella* species set up a solid foundation for further characterizations of this family of effectors. However, future studies will be needed to identify which substrates are physiologically important under infection conditions.

## Adherence to community standards

This study was reported in accordance with the MDAR (Materials Design Analysis Reporting) Framework for life sciences research. A completed MDAR checklist has been provided with the manuscript. No additional specialized reporting guidelines (e.g. CONSORT, PRISMA, ARRIVE, STRANGE) were applicable to this study.

## Materials and methods
### Antibodies and nanobeads

Anti-GFP antibody (polyclonal, source organism rabbit) was generated in-house and a dilution of 1:5000 was used for immunoblot. Anti-GFP nanobeads were generated via amine-coupling reaction of purified recombinant anti-GFP nanobody to Affi-Gel 10 (N-hydroxy-succinimide, Bio-Rad Cat# 1536099) per manufacturer's protocol. Anti-β-actin antibody (monoclonal, source organism mouse) was purchased from Proteintech (Cat# 66009-1-Ig) and a dilution of 1:3000 was used for immunoblot.

Anti-Flag antibody (monoclonal, source organism mouse) was purchased from Sigma-Aldrich (Cat# F3165, 0.2 mg) and a dilution of 1:3000 was used for immunoblot. Anti-Flag beads were purchased from Sigma-Aldrich (Cat# M8823-1ML) and used as per manufacturer's instructions.

### Cell lines

HEK293T cells (source organism human) were purchased from ATCC Cat# CRL-3216. HeLa cells (source organism human) were purchased from ATCC Cat# CCL-2. Cell lines were routinely tested and found negative for mycoplasma contamination.

### Microbes

NEB Stable Competent *Escherichia coli* (High Efficiency) (NEB Cat# C3040I) were used for cloning and mutagenesis. Rosetta DE3 competent cells (Novagen Cat# 70954-3) were used for protein expression.

### Sample definition and in-laboratory replication

Quantitative assays were performed in technical triplicate.

### Attrition

No pre-established exclusion criteria were applied, and no samples or data points were omitted from the analysis.

### Statistics

No inferential statistical tests were performed. Data represent technical triplicates, and variability is reported as standard deviation (SD), calculated using Microsoft Excel. Standard deviation was used to describe measurement variability among technical replicates.

### Plasmid construction

Please refer to *Supplementary file 1* for constructs used in this study. All targeted DNA fragments were amplified by PCR using the primers listed in *Supplementary file 2*. The amplified DNA fragments were digested by restriction enzymes, BamHI and XhoI, and ligated to corresponding vector plasmids. Plasmids containing the targeted genes were amplified using NEB Stable Competent C3040I *E. coli* strain and verified by sequencing. For mutagenesis, plasmids containing WT genes were used as templates with mutagenic primers in a PrimeStar MasterMix PCR to amplify the entire plasmid, followed by DpnI digestion, and plasmids carrying the desired mutations were amplified using the C3040I *E. coli* strain and verified by sequencing.

### Protein purification

Actin was purified from muscle acetone powder prepared from ground beef (*Pardee and Spudich, 1982*). Briefly, 10 g of muscle acetone powder was dissolved with 200 ml G-actin buffer (5 mM Tris pH 7.5, 0.2 mM CaCl$_2$, 0.2 mM ATP, 0.5 mM DTT) at 4°C with stirring for 30 min followed by low-speed centrifugation at 5000–10,000×$g$ for 10–20 min to remove insoluble solids. Soluble extracts were centrifuged at 15,000×$g$ for 60 min at 4°C, and actin was precipitated by the slow addition of solid ammonium sulfate to reach 25% saturation. Precipitates were collected by centrifugation at 15,000×$g$ for 30 min and dissolved in 50 ml G-actin buffer and dialyzed against 2 l of G-actin buffer overnight with two changes of buffer. The dialyzed actin solution was centrifuged at 32,000×$g$ for 60 min to remove any aggregates. Actin was polymerized by the addition of 10× polymerization buffer to a final solution containing 150 mM KCl, 2 mM MgCl$_2$, 2 mM EGTA, and 1 mM ATP. Polymerization was allowed to proceed for 30 min at 25°C, then in a cold room for 90 min. F-actin was pelleted by centrifuging at 100,000×$g$ for 30 min, and the pellet was resuspended in 10–20 ml G-actin buffer and homogenized in a glass-glass homogenizer on ice. The suspension was dialyzed against 2 l of G-actin buffer for 48 hr with four changes of dialysis solution to completely depolymerize the F-actin. The solution was further clarified by centrifugation at 32,000×$g$ for 60 min. The clarified supernatant was loaded onto a Sephadex G-150 or Sephacryl S-100 HR column, equilibrated in G-actin buffer, to remove the remaining contaminating G-ABPs. Peak fractions were analyzed by SDS-PAGE, pooled, and flash-frozen in liquid nitrogen before storage.

Recombinant Lfat1 ABD (WT and mutants) proteins were expressed in Rosetta *E. coli*. The expression was induced by 0.1 mM IPTG at 18°C overnight. The bacteria were harvested by centrifugation at 4000 rpm using the Beckman Coulter JLA-9.1000 rotor for 20 min. The cell pellets were lysed in Buffer A (20 mM Tris pH 7.5, 150 mM NaCl) using sonication. The whole-cell lysate was then centrifuged at 16,000 rpm using the Beckman Coulter JA-25.50 rotor for 40 min at 4°C, and the clarified lysate was then bound to cobalt resins on a rotator at 4°C for 2 hr. The resins were then extensively washed with Buffer A, and the ABD proteins were released from the resin by a SUMO-specific protease, Ulp1. The released proteins were collected and further purified using size-exclusion chromatography on a Superdex S75 column. The purified ABD proteins were concentrated and then flash-frozen in liquid nitrogen for storage.

## Cell culture and co-IP

HEK293T and HeLa cells were maintained at a low passage and grown in Dulbecco's modified Eagle medium (DMEM) supplemented with 10% fetal bovine sera. For transfection, plasmids were mixed with 1 mg/ml polyethylenimine (PEI, MW10K, Millipore Sigma) at a 5:1 ratio in DMEM incubated for 15 min at room temperature and added directly to cells. After 24 hr of transfection, the cells were used either for IP or imaging experiments.

For IP, HEK293T cells transfected with the indicated plasmids after 24 hr were chilled on ice, washed with ice-cold PBS, and detached using lysis buffer (20 mM Tris pH 7.5, 150 mM NaCl, 1 mM DTT, 0.5% Triton X-100, 0.1% sodium deoxycholate, 1 mM PMSF, with Roche protease inhibitor cocktail). The cells were then lysed using a sonicator on ice. The lysate was clarified by centrifugation at 16,000 rpm using the Beckman Coulter JA-25.50 rotor for 15 min at 4°C. The lysate was then incubated with anti-GFP nanobody beads for 2 hr with rotating. The mixture was then washed in a buffer containing 20 mM Tris pH 7.5, 150 mM NaCl, 1 mM DTT, and 0.5% Triton X-100. The washed beads were then dissolved in 1× SDS loading buffer, and the samples were analyzed by SDS-PAGE followed by western blot and scanned using Li-COR Odyssey CLx scanner.

## Fluorescence microscopy

HeLa cells grown on glass cover slides were transfected with the indicated plasmids after 24 hr. The cells were then washed with PBS and fixed in 4% PFA in PBS for 15 min at room temperature. The fixed cells were washed twice with PBS and incubated with Odyssey blocking buffer with 0.1% saponin and rhodamine- or CF647-phalloidin (Thermo Fisher) for 1 hr at room temperature. The stained cells were washed three times with PBS and mounted on a glass specimen slide with Fluoromount-G (Thermo Fisher) and were imaged using a 3i spinning-disc confocal fluorescence microscope. Line-scan analysis was performed using ImageJ (*Schneider et al., 2012*).

## Cryo-EM sample preparation, data collection, data processing

Purified G-actin (1 mg/ml) was mixed with an equal molar of ABD in G-actin buffer and incubated overnight at 4°C. The ABD-F-actin complex samples with a serial dilution were applied to a glow-discharged copper Quantifoil r1.2/1.3 grids and rapidly plunge-frozen in liquid ethane using FEI-Vitrobot-Mark-IV. The vitrified grids were then transferred to liquid nitrogen for storage and data collection.

For data collection, the grids were imaged using a Thermo Fisher Talos Arctica 200 kV electron microscope with a K3 direct electron detector and a Gatan bioquantum energy filter, and the dataset was collected using SerialEM software using the following parameters: –0.4 to –3.0 µm defocus range, pixel size of 1.5879062 Å, total electron dose of 40.68 electrons per Å$^2$, exposure time of 1.23 s, 50 frames per movie, and 4548 total movies. The movies were then imported into CryoSPARC (*Punjani et al., 2017*) for motion correction and patch-CTF estimation. 2173 out of 4548 micrographs were selected after manual curation. The CryoSPARC Helical Tracer job was utilized to perform particle picking with a minimum particle diameter of 50 Å and a maximum diameter of 100 Å, with each particle separated by 100 Å. A total of 1,733,108 particles were extracted from micrographs using a box size of 480 pixels. Iterative 2D classifications were performed on these particles, and low-resolution classes were discarded after each iteration. Finally, 1,220,462 particles were used for ab initio initial model building, followed by iterative helical and local refinements. The final map was

sharpened using CryoSPARC's Sharpening Tool using half-maps and a B-factor of 58.87 as obtained from the Guinier plot.

## Model building, refinement, validation

The atomic model of F-actin (PDB: 7BTI) and the AlphaFold-predicted model of Lfat1ABD were docked into the finalized Cryo-EM density using ChimeraX 1.25 (*Pettersen et al., 2021*). The atomic model of the F-actin-ABD complex was then refined iteratively using Phenix (*Adams et al., 2010*), and the final model, Cryo-EM full-map and half-maps, was validated by the wwPDB.

## F-actin co-sedimentation assay

Recombinant Lfat1 ABD was mixed with buffer control or G-actin at a final concentration of 20 µM. The mixed samples were either incubated in G-actin buffer or F-actin polymerization buffer (50 mM KCl, 2 mM MgCl$_2$) at room temperature for 30 min followed by centrifugation at 70,000 rpm using the Beckman Coulter TLA-100.3 rotor for 30 min at 4°C. The supernatant and pellet were then analyzed on SDS-PAGE.

To determine the F-actin-binding affinity of ABD, purified recombinant Lfat1 ABD WT and mutant proteins were diluted in G-actin buffer in a series of concentrations at 0, 1.25, 2.5, 5, 7.5, 10, 15, 20, 30, 40, 50, 60 µM and were incubated with 20 µM of G-actin. Actin polymerization was initiated at room temperature for 30 min by adding F-actin polymerization buffer. The samples were then centrifuged at 70,000 rpm using the Beckman Coulter TLA-100.3 rotor for 30 min at 4°C, and the supernatants and pellets were then analyzed on SDS-PAGE and staining Coomassie Brilliant Blue dye, and the intensity of the bands was quantified using ImageJ. The intensity of the ABD band was divided by the intensity of the actin band in the pellet fractions to calculate the % of ABD bound to F-actin. The average of each data point from three technical replicates was then plotted, and an exponential fit was used to calculate the apparent $K_d$ in RStudio.

## Protein fatty acylation detection via click chemistry

Indicated plasmids were transfected into HEK 293T cells using PEI transfection reagent. After overnight transfection, cells were treated with 50 µM Alk14 (Cayman Chemical) for 6 hr. The cells were washed with ice-cold PBS and then lysed in lysis buffer (25 mM Tris-HCl, pH 7.8, 150 mM NaCl, 10% glycerol, and 1% NP-40) with protease inhibitor cocktail at 4°C for 30 min. After centrifugation at 17,000×$g$ for 30 min at 4°C, the supernatant was collected and incubated with 20 µl of anti-Flag affinity beads (Sigma-Aldrich) at 4°C for 2 hr. The affinity beads were washed three times with washing buffer (25 mM Tris-HCl, pH 7.8, 150 mM NaCl, 0.2% NP-40) and resuspended in 20 µl of washing buffer. TAMRA-N3 (Lumiprobe), TBTA (TCI Chemicals), CuSO$_4$, and TCEP (Millipore) were added into the reaction mixture in the order listed. The click chemistry reaction was allowed to proceed at room temperature for 30 min. The reaction was quenched by adding 6× SDS loading dye and boiled for 5 min. Where indicated, samples were treated with hydroxylamine to remove cysteine fatty acylation. The samples were then separated on SDS-PAGE and fixed in a buffer (50% CH$_3$OH, 40% water, and 10% acetic acid) by shaking for 1 hr at 4°C and then washed and stored in water. The gel was scanned to record fluorescence signal using a ChemiDoc MP (Bio-Rad).

## SILAC sample preparation and MS data analysis

SILAC samples were prepared from cells transiently expressing WT or H38A mutant Lfat1 using a published protocol (*Kosciuk et al., 2020*). The samples were then trypsin-digested, and the peptides were analyzed using an Orbitrap Fusion Tribrid (Thermo Fisher Scientific) mass spectrometer. The MS and MS/MS spectra were subjected to database searches using Proteome Discoverer (PD) 2.4 software (Thermo Fisher Scientific, Bremen, Germany) with the Sequest HT algorithm. The database search was conducted against a *Homo sapiens* Uniprot database with the following variable modifications: methionine oxidation; deamidation of asparagine/glutamine; SILAC heavy: R10 (10.008 Da) and K8 (8.014 Da) and light labeling on R and K; palmitoylation plus biotin on K, protein N-terminus, and fixed modification of cysteine carbamidomethylation. Full list of hits is in *Supplementary file 3* (forward-SILAC) and *Supplementary file 4* (reverse-SILAC).

## Materials availability statement

All materials generated in this study, including antibodies (in-house), nanobeads, plasmids, and protein constructs, are available from the corresponding author upon reasonable request. Distribution of materials may be subject to a material transfer agreement and institutional policies.

## Acknowledgements

We thank Dr. Sheng Zhang and Dr. Qin Fu for helping with the proteomics study. WZ acknowledges support from the T32 training grant and the Sadov Graduate Student Fellowship (Cornell University). This work is supported by the National Institutes of Health R01GM144452 (YM), R01AI153110 (HL), and HHMI.

## Additional information

### Funding

| Funder | Grant reference number | Author |
|---|---|---|
| National Institutes of Health | T32 | Wenjie W Zeng |
| Cornell University | the Sadov Graduate Student Fellowship | Wenjie W Zeng |
| National Institutes of Health | R01GM144452 | Yuxin Mao |
| National Institutes of Health | R01AI153110 | Hening Lin |
| Howard Hughes Medical Institute | | Hening Lin |

The funders had no role in study design, data collection and interpretation, or the decision to submit the work for publication.

### Author contributions

Wenjie W Zeng, Conceptualization, Data curation, Formal analysis, Investigation, Methodology, Writing – original draft, Writing – review and editing; Garrison Komaniecki, Data curation, Formal analysis, Investigation, Methodology; Jiaze Liu, Data curation, Investigation, Methodology; Hening Lin, Formal analysis, Funding acquisition, Investigation; Yuxin Mao, Conceptualization, Data curation, Formal analysis, Supervision, Funding acquisition, Investigation, Writing – original draft, Project administration, Writing – review and editing

### Author ORCIDs

Wenjie W Zeng ⓘ https://orcid.org/0000-0003-1028-7845
Jiaze Liu ⓘ https://orcid.org/0009-0002-9962-8086
Hening Lin ⓘ https://orcid.org/0000-0002-0255-2701
Yuxin Mao ⓘ https://orcid.org/0000-0002-5064-1397

Reviewer #1 (Public review): https://doi.org/10.7554/eLife.106975.3.sa1
Reviewer #2 (Public review): https://doi.org/10.7554/eLife.106975.3.sa2
Author response https://doi.org/10.7554/eLife.106975.3.sa3

## Additional files

### Supplementary files

Supplementary file 1. List of constructs used in this study.
Supplementary file 2. List of primers used in this study.

Supplementary file 3. Forward Stable Isotope Labeling by Amino Acids in Cell Culture (SILAC) mass spectrometry (MS) result.

Supplementary file 4. Reverse Stable Isotope Labeling by Amino Acids in Cell Culture (SILAC) mass spectrometry (MS) result.

MDAR checklist

## Data availability
Structural coordinates were deposited at RSCB with the access code: 8VAA. Cryo-EM Map was deposited at EMDB with a code: 43087.

The following datasets were generated:

| Author(s) | Year | Dataset title | Dataset URL | Database and Identifier |
| --- | --- | --- | --- | --- |
| Zeng W, Mao Y | 2024 | Actin-binding domain of Legionella pneumophila effector LFAT1 (lpg1387) bound to F-actin | https://doi.org/10.2210/pdb8VAA/pdb | Worldwide Protein Data Bank, 10.2210/pdb8VAA/pdb |
| Zeng W, Mao Y | 2024 | Actin-binding domain of Legionella pneumophila effector LFAT1 lpg1387 bound to F-actin | https://www.ebi.ac.uk/emdb/EMD-43087 | Electron Microscopy Data bank, EMD-43087 |

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
