## [Editor Report · eLife Assessment]

This **important** study by Zeng et al characterizes a novel Legionella pneumophila effector, Llfat1 (Lpg1387), which binds actin through a newly identified actin-binding domain. Data is **convincing**; structural analysis of the Llfat1 ABD-F-actin complex enabled the development of this domain as a probe for F-actin. Additionally, the authors show that Llfat1 functions as a lysine fatty acyltransferase targeting small GTPases, highlighting its importance in both bacterial pathogenesis and cytoskeletal biology.

---

## [Referee Report · Reviewer #1 (Public review)]

The manuscript by Zeng et al. describes the discovery of an F-actin-binding Legionella pneumophila effector, which they term Lfat1. Lfat1 contains a putative fatty acyltransferase domain that structurally resembles the Rho-GTPase Inactivation (RID) domain toxin from Vibrio vulnificus, which targets small G-proteins. Additionally, Lfat1 contains a coiled-coil (CC) domain.

The authors identified Lfat1 as an actin-associated protein by screening more than 300 Legionella effectors, expressed as GFP-fusion proteins, for their co-localization with actin in HeLa cells. Actin binding is mediated by the CC domain, which specifically binds to F-actin in a 1:1 stoichiometry. Using cryo-EM, the authors determined a high-quality structure of F-actin filaments bound to the actin-binding domain (ABD) of Lfat1. The structure reveals that actin binding is mediated through a hydrophobic helical hairpin within the ABD (residues 213-279). A Y240A mutation within this region increases the apparent dissociation constant by two orders of magnitude, indicating a critical role for this residue in actin interaction.

The ABD alone was also shown to strongly associate with F-actin upon overexpression in cells. The authors used a truncated version of the Lfat1 ABD to engineer an F-actin-binding probe, which can be used in a split form. Finally, they demonstrate that full-length Lfat1, when overexpressed in cells, fatty acylates host small G-proteins, likely on lysine residues.

Comments on revisions:

Since LFAT1 cannot be produced in *E. coli*, it may be worth considering immunoprecipitating the protein from mammalian cells to see if it has activity in vitro. Presumably, actin will co-IP but the actin binding mutant can also be used. These are just suggestions to improve an already solid manuscript. Otherwise, I am happy with the paper.

---

## [Referee Report · Reviewer #2 (Public review)]

Summary:

The manuscript by Zeng et al reports the structural and biochemical study of a novel effectors from the bacterial pathogen Legionella pneumophila. The authors continued from results from their earlier screening for L. pneumophila proteins that that affect host F-actin dynamics to show that Llfat1 (Lpg1387) interacts with actin via a novel actin-binding domain (ABD). The authors also determined the structure of the Lfat1 ABD-F-actin complex, which allowed them to develop this ABD as probe for F-actin. Finally, the authors demonstrated that Llfat1 is a lysine fatty acyltransferase that targets several small GTPases in host cells. Overall, this is a very exciting study and should be of great interest to scientists in both bacterial pathogenesis and actin cytoskeleton of eukaryotic cells.

---

## [Author Response]

The following is the authors’ response to the original reviews

**Reviewer #1:**
(1) Legionella effectors are often activated by binding to eukaryote-specific host factors, including actin. The authors should test the following: (a) whether Lfat1 can fatty acylate small G-proteins in vitro; (b) whether this activity is dependent on actin binding; and (c) whether expression of the Y240A mutant in mammalian cells affects the fatty acylation of Rac3 (Figure 6B), or other small G-proteins.

We were not able to express and purify the full-length recombinant Lfat1 to perform fatty acylation of small GTPases in vitro. However, in cellulo overexpression of the Y240A mutant still retained ability to fatty acylate Rac3 and another small GTPase RheB (see Figure 6-figure supplement 2). We postulate that under infection conditions, actin-binding might be required to fatty acylate certain GTPases due to the small amount of effector proteins that secreted into the host cell.

(2) It should be demonstrated that lysine residues on small G-proteins are indeed targeted by Lfat1. Ideally, the functional consequences of these modifications should also be investigated. For example, does fatty acylation of G-proteins affect GTPase activity or binding to downstream effectors?

We have mutated K178 on RheB and showed that this mutation abolished its fatty acylation by Lfat1 (see Author response image 1 below). We were not able to test if fatty acylation by Lfat1 affect downstream effector binding.

(3) Line 138: Can the authors clarify whether the Lfat1 ABD induces bundling of F-actin filaments or promotes actin oligomerization? Does the Lfat1 ABD form multimers that bring multiple filaments together? If Lfat1 induces actin oligomerization, this effect should be experimentally tested and reported. Additionally, the impact of Lfat1 binding on actin filament stability should be assessed. This is particularly important given the proposed use of the ABD as an actin probe.

The ABD domain does not form oligomer as evidenced by gel filtration profile of the ABD domain. However, we do see F-actin bundling in our in vitro -F-actin polymerization experiment when both actin and ABD are in high concentration (data not shown). Under low concentration of ABD, there is not aggregation/bundling effect of F-actin.

(4) Line 180: I think it's too premature to refer to the interaction as having "high specificity and affinity." We really don't know what else it's binding to.

We have revised the text and reworded the sentence by removing "high specificity and affinity."

(5) The authors should reconsider the color scheme used in the structural figures, particularly in Figures 2D and S4.

Not sure the comments on the color scheme of the structure figures.

(6) In Figure 3E, the WT curve fits the data poorly, possibly because the actin concentration exceeds the Kd of the interaction. It might fit better to a quadratic.

We have performed quadratic fitting and replaced Figure 3E.

(7) The authors propose that the individual helices of the Lfat1 ABD could be expressed on separate proteins and used to target multi-component biological complexes to F-actin by genetically fusing each component to a split alpha-helix. This is an intriguing idea, but it should be tested as a proof of concept to support its feasibility and potential utility.

It is a good suggestion. We plan to thoroughly test the feasibility of this idea as one of our future directions.

(8) The plot in Figure S2D appears cropped on the X-axis or was generated from a ~2× binned map rather than the deposited one (pixel size ~0.83 Å, plot suggests ~1.6 Å). The reported pixel size is inconsistent between the Methods and Table 1-please clarify whether 0.83 Å refers to super-resolution.

Yes, 0.83 Å is super-resolution. We have updated in the cryoEM table

**Reviewer #2:**
Weaknesses:(1) The authors should use biochemical reactions to analyze the KFAT of Llfat1 on one or two small GTPases shown to be modified by this effector in cellulo. Such reactions may allow them to determine the role of actin binding in its biochemical activity. This notion is particularly relevant in light of recent studies that actin is a co-factor for the activity of LnaB and Ceg14 (PMID: 39009586; PMID: 38776962; PMID: 40394005). In addition, the study should be discussed in the context of these recent findings on the role of actin in the activity of L. pneumophila effectors.

We have new data showed that Actin binding does not affect Lfat1 enzymatic activity. (see response to Reviewer #1). We have added this new data as Figure S7 to the paper. Accordingly, we also revised the discussion by adding the following paragraph.

“The discovery of Lfat1 as an F-actin–binding lysine fatty acyl transferase raised the intriguing question of whether its enzymatic activity depends on F-actin binding. Recent studies have shown that other *Legionella* effectors, such as LnaB and Ceg14, use actin as a co-factor to regulate their activities. For instance, LnaB binds monomeric G-actin to enhance its phosphoryl-AMPylase activity toward phosphorylated residues, resulting in unique ADPylation modifications in host proteins (Fu et al, 2024; Wang et al, 2024). Similarly, Ceg14 is activated by host actin to convert ATP and dATP into adenosine and deoxyadenosine monophosphate, thereby modulating ATP levels in *L. pneumophila*–infected cells (He et al, 2025). However, this does not appear to be the case for Lfat1. We found that Lfat1 mutants defective in F-actin binding retained the ability to modify host small GTPases when expressed in cells (Figure S7). These findings suggest that, rather than serving as a co-factor, F-actin may serve to localize Lfat1 via its actin-binding domain (ABD), thereby confining its activity to regions enriched in F-actin and enabling spatial specificity in the modification of host targets.”

(2) The development of the ABD domain of Llfat1 as an F-actin domain is a nice extension of the biochemical and structural experiments. The authors need to compare the new probe to those currently commonly used ones, such as Lifeact, in labeling of the actin cytoskeleton structure.

We fully agree with the reviewer’s insightful suggestion. However, a direct comparison of the Lfat1 ABD domain with commonly used actin probes such as Lifeact, as well as evaluation of the split α-helix probe (as suggested by Reviewer #1), would require extensive and technically demanding experiments. These are important directions that we plan to pursue in future studies.

For all other minors, we have made corrections/changes in our revised text and figures.